# Redox Chemistry: Implications for Necrotizing Enterocolitis

**DOI:** 10.3390/ijms25158416

**Published:** 2024-08-01

**Authors:** Grant H. Gershner, Catherine J. Hunter

**Affiliations:** 1Division of Pediatric Surgery, Oklahoma Children’s Hospital, 1200 Everett Drive, ET NP 2320, Oklahoma City, OK 73104, USA; grant-gershner@ouhsc.edu; 2Department of Surgery, The University of Oklahoma Health Sciences Center, 800 Research Parkway, Suite 449, Oklahoma City, OK 73104, USA

**Keywords:** oxidative stress, necrotizing enterocolitis, neonate, prematurity, reactive oxygen species, reactive nitrogen species

## Abstract

Reduction–oxidation (redox) chemistry plays a vital role in human homeostasis. These reactions play critical roles in energy generation, as part of innate immunity, and in the generation of secondary messengers with various functions such as cell cycle progression or the release of neurotransmitters. Despite this cornerstone role, if left unchecked, the body can overproduce reactive oxygen species (ROS) or reactive nitrogen species (RNS). When these overwhelm endogenous antioxidant systems, oxidative stress (OS) occurs. In neonates, OS has been associated with retinopathy of prematurity (ROP), leukomalacia, and bronchopulmonary dysplasia (BPD). Given its broad spectrum of effects, research has started to examine whether OS plays a role in necrotizing enterocolitis (NEC). In this paper, we will discuss the basics of redox chemistry and how the human body keeps these in check. We will then discuss what happens when these go awry, focusing mostly on NEC in neonates.

## 1. Introduction

Necrotizing enterocolitis (NEC) has been well described as one of the most devastating diseases in the neonatal period. NEC is a gastrointestinal disease of mostly premature neonates that primarily affects the small intestine. The areas most commonly affected include the jejunum and ileum, although the colon can also be involved [1]. NEC initially presents with early signs of infection including temperature instability, apnea, and lethargy. This may be accompanied by feeding intolerance, abdominal distension, or bloody bowel movements. As NEC progresses clinically, the patient will become more unstable and require the initiation of vasoactive medications. Left untreated, this will lead to multi-organ failure and the patient’s demise [2]. Radiographically, NEC is characterized by pneumatosis intestinalis (air accumulation within the wall of the bowel). In some cases, this air is absorbed into the venous system and can accumulate in the liver, leading to portal venous gas accumulation. Statistically, NEC has a prevalence of 7% and an associated morbidity of between 20 and 30%. It has a higher prevalence among premature neonates and those with low birth weights (500–1500 g) [3]. While research has been ongoing, relatively few improvements have been made over the years, and it continues to plague neonatal intensive care units across the world. NEC is a multifactorial phenomenon with potential etiologies including an immature gut barrier, an incomplete gut microbiome, hypoxia, metabolite overload from feeding exposure (especially with formula), apoptosis sensitivity, and bacterial overload [4]. Oxidative stress (OS) has been shown to result in long-term morbidity in infants in the form of retinopathy of prematurity, bronchopulmonary dysplasia, periventricular leukomalacia, and more [5]. The birthing process is known to be an oxidatively stressful event [6]. This stressful event is compounded with subsequent high concentrations of oxygen exposure (due to respiratory distress or reperfusion injury) in the setting of hypoxia. With OS causing multi-organ pathologies, researchers are beginning to examine the possibility that OS might also contribute to the pathology of NEC. In this paper, we will summarize our current understanding of redox chemistry and how it contributes to oxidative stress as well as how it affects the endogenous systems that help defend the body. We will also examine how these systems can be overloaded in the neonatal period and how this can lead to increased morbidity and an increased susceptibility to NEC. We will look at the current diagnostic approaches and therapeutic strategies being used. Finally, we will discuss emerging technologies in the field of Redox Biology and their implications for NEC treatment.

## 2. Necrotizing Enterocolitis

NEC is a gastrointestinal disease of neonates characterized by inflammation, necrosis, and potential perforation of the intestinal tissue. NEC has an incidence of 0.3 to 2.4 infants per 1000 live births. Of those, 70% of cases are in infants born at less than 36 weeks gestation [7]. The mortality of this disease varies based on the stage of NEC (23.5%, Bell Stage IIa), need for surgery (34.5% in normal-weight, 50.9% in extremely-low-birth-weight surgical patients), and birth weight (40.5% in extremely-low-birth-weight patients) [8]. It is associated with neurodevelopmental disabilities (24.8–61.1%) and syndromes of intestinal failure like short gut syndrome (15.2–35.0%) [8].

Research thus far has found NEC to be a multifactorial process. Current known risk factors include immaturity of the gastrointestinal tract, altered gut microbiota, and compromised immune responses in neonates [4]. Prematurity is also a prominent known risk factor. This is in part due to the underdeveloped intestinal mucosa, which makes infants more susceptible to injury and inflammation. Early feeds, specifically formula feeding, also increase the risk of NEC. While instances of NEC do happen in breastfed infants, studies have shown early exposure to formula feeds greatly increases the risk compared to in breastfed counterparts [9]. Additionally, patients with NEC also exhibit a hyperinflammatory state from exposure to the above risks. This inflammation subsequently leads to the “leaky gut”, or impaired intestinal barrier function [10]. This allows for luminal contents to enter the intestinal wall and bloodstream, leading to bowel compromise and sepsis.

Pro-inflammatory cytokines and chemokines serve as crucial early-warning biomarkers for diagnosing NEC. These molecules, released in response to intestinal injury and inflammation, play a pivotal role in the pathogenesis of NEC by promoting and sustaining the inflammatory cascade. Elevated levels of cytokines such as tumor necrosis factor-alpha (TNF-α) and interleukins (IL-6, IL-8) have been observed in neonates with NEC, indicating their potential utility in early detection [11]. Similarly, chemokines like monocyte chemoattractant protein-1 (MCP-1) contribute to the recruitment of immune cells to the site of injury, exacerbating the inflammatory response [12]. Many of these inflammatory cytokines are stimulated by TLR4 activation [13]. Lipopolysaccharide (LPS), a commonly found component of Gram-negative bacterial walls, directly stimulates TLR4, resulting in intestinal inflammation [14]. Recently, Cho et al. found that, not only were TLR4 levels elevated, but so were levels of TLR8. Conversely, levels of TLR5–7 and TLR9–12 were reduced [15].

Currently, NEC is diagnosed clinically, relying on the clinician’s judgment to integrate physical examination findings, standard laboratory values, and imaging results. As previously mentioned, common clinical signs include temperature irregularity, apnea, feeding intolerance, abdominal distension, bloody bowel movements, and hemodynamic instability. The standard imaging used to diagnose NEC is an anteroposterior (AP) and lateral X-ray. This can illustrate pneumatosis intestinalis, portal venous gas, or free air in instances of perforation. Laboratory values will usually show leukocytosis, thrombocytopenia, and elevated lactic acid. Several studies have sought a biomarker to aid in the diagnosis and staging of NEC. These include apolipoproteins, intestinal alkaline phosphatase, miRNA, fecal calprotectin, and claudins. However, none of these biomarkers has yet proven to be significantly effective [16].

Treatment of NEC is determined by severity. NEC severity is classified using Bell’s staging system. This is a three-stage system. Stage I (Mild/Suspected NEC) is defined as mild NEC with mild systemic and intestinal signs. Treatment for this is close clinical observation and holding of enteral feeds. Stage II has prominent abdominal distension and abdominal tenderness. Thrombocytopenia can develop on complete blood counts (CBC) along with leukocytosis. At this stage, you can usually see the imaging findings previously listed. This stage is managed with nasogastric decompression, fluid resuscitation, and broad-spectrum antibiotics. Finally, Stage III is advanced NEC. The patient is usually peritonitic on abdominal exam and will have severe metabolic acidosis on labs. If the gastrointestinal viscus has perforated, then free air will be seen on imaging. If the child is large enough, surgery with exploratory laparotomy and bowel resection is indicated. If the child is not large enough (<750 g), then peritoneal drainage is indicated [17].

## 3. Redox Chemistry

At its core, redox reactions involve the transfer of electrons from a reducing agent to an oxidizing agent. The reducing agent donates electrons (undergoing oxidation), while the oxidizing agent accepts electrons (undergoing reduction) [18]. This fundamental interplay has many essential roles in cellular functions including energy production, metabolism, innate immunity, and cell signaling.

### 3.1. Reactive Oxygen Species

Reactive oxygen species are highly reactive molecules containing oxygen, such as superoxide radicals, hydrogen peroxide, and hydroxyl radicals [19]. While ROS are produced naturally as byproducts of cellular metabolism (Figure 1), they can also be generated in response to external stimuli such as UV radiation, pollutants, and toxins. Excessive ROS production or inadequate antioxidant defense can lead to oxidative stress, causing damage to lipids, proteins, and DNA. A list of common ROS can be found in Table 1.

### 3.2. Reactive Nitrogen Species

Similar to ROS are reactive nitrogen species (RNS) [20]. These are similarly highly reactive molecules but with nitrogen at their base instead of oxygen. The most well known of these is nitric oxide (NO) and its derivatives. NO has many significant physiological functions. It mainly acts as a signaling molecule to induce vasodilation, neurotransmitter release, and immune response regulation [21]. NO has several byproducts that act as RNS, such as peroxynitrite and nitrogen dioxide [20].

## 4. Physiologic Redox Chemistry

### 4.1. Reactive Species Sources and Functions

Reactive species (RS) are generally formed in two ways. The first way is when they are produced naturally within the cell through normal cellular processes. These are considered to be intrinsic RS. In contrast, extrinsic RS are those that are made when other factors encourage RS formation such as radiation, pollutants, metals, toxin exposure, and bacterial invasion [22,23].

ROS are generally produced due to metabolic stress or as a part of cellular homeostasis. A primary example of physiologic redox chemistry occurs in the mitochondria. Using the electron transport chain (ETC), mitochondria generate superoxide radicals and hydrogen peroxide in almost every complex [24]. Redox chemistry is found in hormonal control as well. When obese patients underwent fasting, they were found to have a more oxidized state compared to lean patients [25]. This oxidized state often includes higher levels of H_2_O_2_. Research has demonstrated that H_2_O_2_ can directly stimulate insulin secretion, independent of glucose concentration [26]. Similarly, ROS are used by the innate immune system as a form of biocide [27]. Within macrophage peroxisomes, NADPH Oxidases (NOXs) generated an oxidative burst, a burst of reactive species (RS) that destroyed phagocytized pathogens [28]. NOXs also play a pivotal role in cellular signaling. After activation by hormones, like insulin and interferon-gamma (IFN-γ), NOXs form H_2_O_2_, leading to intracellular oxidation and subsequent protein modification through the oxidation of cysteine residues [29]. Different proteins react differently to oxidation. Generally, phosphatases are inhibited, while kinases are more varied in their response [29]. ROS are also made through other enzymatic reactions that include xanthine oxidase, catalase, superoxide dismuates, and others [30].

The main source of RNS is NO synthases (NOSs). These convert L-arginine and oxygen into NO [31]. There are three isoforms of this enzyme: neuronal (nNOS), inducible (iNOS), and endothelial (eNOS). nNOS is constitutively activated and is used to synthesize moderate amounts of NO in nerves [31]. Similarly, eNOS is constitutively activated and releases NOs in endothelial cells to induce vasodilation [32].

In terms of DNA regulation, redox factors can control transcription factors (TFs). Notable TFs affected by redox status are nuclear factor erythroid 2-related factor 2 (NRF2), nuclear factor kappa B (NF-κB), hypoxia-inducible factor 1 alpha (HIF1α), and forkhead box O (FOXO) [29]. Under homeostatic conditions, many of these factors are degraded or are unable to cross the nuclear membrane. However, upon the development of OS, these factors can enter the nucleus and influence cellular epigenetics.

### 4.2. Enzymatic and Non-Enzymatic Regulation

While these RS play critical roles in normal physiology, they must be tightly regulated to prevent inadvertent damage. The human body has evolved several innate antioxidant systems to keep this process in check. There are several families of enzymes that are the main defense against OS. One such example is superoxide dismutase, which breaks down superoxide radicals into oxygen and hydrogen peroxide. This is followed by catalase, which can further process the hydrogen peroxide into water and oxygen. Another family is the glutathione (GSH) and glutathione peroxidase (GPX) system. GSH is a tripeptide made of cysteine, glycine, and glutamate. It exists in two forms, GSH (reduced) and GSSG (oxidized). In its reduced form, it is a strong scavenger of both ROS and RNS. Once oxidized, it is used as a cofactor for GPXs. This recycles GSSG back into GSH, allowing it to continue scavenging RS. The GPX system is also known for using selenoproteins. Of the six known GPX enzymes, five are selenocysteine containing (GPX 1, 2, 3, 4, 6) [33]. In a typical redox cycle, the Sec is oxidized to selenic acid, which is subsequently reduced by GSH [33] (Figure 2).

The thioredoxin (Trx) system is a family of proteins and enzymes (similar to the GSH system) that aid in the degradation of RS [34]. Using Thioredoxin reductase (TrxR), Trx undergoes reduction of active site disulfide bonds with NADPH to help recycle it back into the system. Trx is used as a general reductant for protein disulfides as well as several other low-molecular-weight compounds [35]. Like the GPX system, Trx is a selenoprotein. For Trx, this is found in the C-terminal active site, which helps it form a redox-active center [33].

Similar to enzymatic antioxidants, several non-enzymatic antioxidants aid in the regulation of RS. Some examples are vitamins. The vitamin C family is one of these, functioning as a scavenger for both RS generated intrinsically and extrinsically [36]. Vitamin E is also a potent antioxidant, especially in cell membranes [36]. Carotenoids, derived from orange, red, and yellow pigmented plants, can be converted by the body into vitamin A. Carotenoids themselves are excellent scavengers of singlet oxygen and other ROS [36].

If these systems are overwhelmed, then oxidative damage may occur. This oxidative damage contributes to the pathogenesis of various diseases, including neurodegenerative disorders, cardiovascular diseases, and cancer [37,38,39]. Damage occurs through two pathways. The first is through the direct oxidation of macromolecules (lipids, proteins, nucleic acids, etc.) by RS. The other is through aberrant redox signaling. Both of these pathways lead to subsequent deterioration in physiologic activity. Therefore, maintaining the delicate balance of ROS levels is essential for cellular homeostasis and overall health.

## 5. Redox Chemistry and Necrotizing Enterocolitis 

### 5.1. Implications of Redox Chemistry in Necrotizing Enterocolitis

The pathogenesis of NEC starts with an initial trigger. This initial stress response causes a release of inflammatory mediators. Compounds like platelet-activating factor (PAF) [40], TNFα, and various interleukins [41] initiate the inflammatory cascade. Once activated, polymorphonuclear leukocytes (PMNs) migrate into the bowel wall [42]. Activation of PMNs and other immune complexes results in further ROS production [43]. The hyperinflammatory state combined with hypoxia, reperfusion injury, and bacterial colonization leads to an abundance of RS.

A link between NEC and OS has been previously suggested. A previous study by Aydemire et al. found that preterm infants with NEC had significantly higher total oxidant status (TOS) and higher oxidative stress indexes (OSI) [44]. This theory was further bolstered when cord blood from preterm neonates was examined by Perrone et al. Their markers of oxidative stress included non-protein-bound iron (NPBI), advanced oxidation protein products (AOPP), and total hydroperoxides (TH). They found that all markers were significantly elevated in babies with NEC [45]. This has been found in surgical specimens as well. Zamora et al. located higher levels of NO and its metabolites in the gut of surgical specimens from neonates with NEC [46]. ROS have also been found to induce the expression of cyclooxygenase 2 (COX2). COX2 is an inducible enzyme that is found to be especially high in times of inflammation [47]. A study by Chung et al. found similarly elevated levels in affected tissue in patients with NEC, while healthy tissue from the same sample had normal levels [48].

Similarly, a link between NEC and Nitrositive stress has been suggested [49]. One mechanism of cellular damage is through NO. NO readily reacts with superoxide (O_2_^−^) to form peroxynitrite (ONOO^−^), a strong oxidant [50]. This form of RNS readily reacts with lipids and proteins, causing nitrosative damage [51,52]. This type of damage is seen in inflammatory intestinal diseases like ulcerative colitis and Crohn’s disease [53]. Ford and colleagues found that levels of iNOS were elevated in neonates with NEC. This elevated iNOS level leads to higher levels of NO with subsequent apoptosis of enterocytes via peroxynitrite formation [54]. The formation of peroxynitrites leads to mitochondrial proteins undergoing nitrosylation, causing the inhibition of cellular respiration and eventual apoptosis activation via the caspase cascade [55]. Finally, elevated levels of NO affect gut barrier function. This occurs due to S-nitrosylation of barrier lipids and proteins [56], contributing to the leaky gut of NEC [10].

Attempts to illustrate this connection with experimental models have been trialed. Clark et al. showed that rabbits exposed to intraluminal oxidative conditions developed intestinal damage and subsequent tissue necrosis. This was completely ameliorated with the infusion of SOD [57]. Nadler et al. found that neonatal rats exposed to formula had increased levels of iNOS regardless of exposure to hypoxia [58]. Finally, Zamora et al. found that neonatal rats exposed to hypoxia and formula feeding had increased levels of NO metabolites as well as upregulation of iNOS in NEC samples, consistent with other models [46].

### 5.2. Risk Factors

In addition to the foregoing associations, neonates are exposed to several risk factors for OS. Those that are preterm or small for gestation age (SGA) have been shown to have increased ROS production [59]. This is in part due to an elevated level of metabolic turnover [60]. Preterm infants can also be exposed to high amounts of concentrated oxygen due to respiratory distress and resuscitative needs. During these resuscitative efforts, 100% oxygen is given [61]. In a sheep model, even three minutes of exposure to 100% oxygen caused a rise in fetal ROS [62].

Nutritional factors contribute to the elevated OS of prematurity. When possible, enteral feeds, especially with mothers’ milk, are the standard. If mothers’ milk is not an option, then formulas are pursued. While formula feeds provide adequate nutrition, they are associated with an elevated risk of NEC [63]. This is in part due to formulas containing higher amounts of iron compared to breast milk. Iron is a strong oxidizer and is known to cause excess production of free radicals [64]. Studies have shown that, with exposure to higher iron levels, there is an increase in FR production, lipid peroxidation, and OS [65,66]. For those who cannot tolerate oral feeds, nutrition is provided through total parenteral nutrition (TPN). While TPN can provide vital nutrition to neonates, it can further lead to OS. TPN solutions that are fat free can contain varying amounts of hydrogen peroxide. The peroxide is formed when vitamin complexes, namely 5′ phosphate flavin mononucleotide (FMN) and polysorbates (PS), react with electron donors when exposed to light [67].

This overproduction of RS can overwhelm the physiologic antioxidant systems. At baseline, newborns are more susceptible to oxidative stress. The womb is a hypoxic environment compared to the extrauterine world. The intrauterine environment typically has an oxygen tension of 20–25 mm Hg PO_2_, while that of the extrauterine air is around 100 mm Hg PO_2_ [68]. Additionally, newborns have been found to have higher levels of free iron. High levels of free iron are known to increase the Fenton Reaction, the process by which ferrous iron reacts with hydrogen peroxide to form Ferric iron, a hydroxyl, and a highly reactive hydroxyl radical [69,70]. Normally, the fetal antioxidant system ramps up in the last 15% of gestation. This is in part due to antioxidants (vitamins, superoxide dismutase, etc.) crossing the placenta [71]. The transfer is critical due to neonatal endogenous antioxidant systems being immature. Normally, the neonatal antioxidant defense systems ramp up activity during the later months to weeks of gestation. Without time for development, premature neonates are further susceptible to OS [72].

## 6. Reducing Oxidative Stress and Necrotizing Enterocolitis Risk

Several methods are being examined to reduce the oxidative stress in neonates. Firstly, breast milk has been shown to be extremely beneficial. Not only does it contain vital nutrition and promote proper neonatal gut health and bacterial colonization, but it also contains natural antioxidants [73]. Studies have shown that both mothers’ milk and human donor breast milk have higher levels of natural antioxidants when compared to formula [74,75]. If formula feedings are necessary, consideration of the nutritional fat profile is warranted. Sodhi et al. found that mice exposed to formulas with standard formula fats had a higher incidence and severity of NEC, while those supplemented with pre-digested fats or a very-low-fat-containing formula had a much lower incidence of NEC [76].

As previously stated, TPN has been shown to carry varying amounts of hydrogen peroxide. To prevent the vitamin complex from reacting, TPN must be properly stored and shielded from light [67]. The composition of TPN fat solutions also warrants examination. Studies have shown that neonates being supplemented with higher levels of short-chain fatty acids (SCFA) led to a higher incidence of NEC [77], while elevated levels of polyunsaturated fatty acids (PUFAs) have been shown to decrease the incidence of NEC due to its immunomodulatory effects [78].

Finally, limiting oxygen exposure to use only when absolutely necessary may help prevent excessive oxygen exposure. Oxygen supplementation is used frequently in the delivery room and perinatal period to help infants transition to life outside the womb. While this stabilization is vital, target oxygen saturations and therapy levels are currently not established [79,80].

## 7. Future Treatments and Directions

Potential therapies have been investigated in the prevention of OS-induced NEC. One potential treatment is melatonin, which is best known for its use in sleep regulation. It has recently been discovered to be an effective scavenger of free radicals due to it being an indirect antioxidant [81,82]. Additionally, supplementation of formulas with lutein has been explored. Thus far, it has failed to show any major impact [83]. More experimentally, Ozone has been trialed as a possible therapeutic for OS. A study by Guven et al. treated a group of rats that underwent NEC induction with medical ozone. Rats in the treated group had lower levels of oxidative stress markers (specifically, malondialdehyde and protein carbonyl content) and more active endogenous antioxidant enzymes (SOD and GPX). They additionally had lower levels of TNFα and improved histologic grading compared to the NEC-only group [84].

With the importance of OS becoming more prominent, new diagnostic tests are being examined to try and quantify oxidative damage or predict severity. One of these markers is urinary 8-hydroxy-2′-deoxyguanosine [85]. This is a byproduct of oxidative DNA damage [86]. Tsukahara et al. found elevated levels of this product in preterm neonates that were clinically sick as compared to healthy preterm neonates [85]. Acrolein–lysine adducts are also being examined as a marker of lipid peroxidation and oxidative protein damage [87]. A more specific marker of lipid peroxidation is F2-isoprostanes. These are a byproduct of lipid peroxidation via radicals that are independent of cyclooxygenase pathways [88]. Markers that can be checked via cord blood samples to see which neonates could be at risk of NEC are another future direction. Potential markers include non-protein-bound iron, advanced oxidation protein products, and total hydroperoxides [45].

Despite promising research into reducing oxidative stress and improving NEC outcomes, a reliable therapeutic target has not been fully identified. Most studies that are investigating this pathway rely on using murine, rat, or pig models. While these allow us to examine the multi-faceted problem of NEC in a scientific and more accessible way, they limit the generalizability to human patients while also having species-specific conflicts [89]. This problem may be mitigated with the recent discovery and study of small bowel organoids, or enteroids [90]. Intestinal crypt stem cells are harvested to generate cell cultures that retain host characteristics, enabling their classification into distinct groups. Furthermore, these cultures can be subjected to NEC-like conditions, such as lipopolysaccharide (LPS) exposure and hypoxia [91], to study cell responses to these stimuli.

Additionally, the short- and long-term side effects of these interventions are yet to be seen. While studying RS themselves is rather difficult given their short half-lives, study into the byproducts of oxidative stress and damage has found several suitable targets. A major challenge will be ensuring the feasibility of running tests such as these in the clinical setting. Further application of these tests to develop a screening test or way to stratify oxidative stress severity (and potentially NEC severity) would provide immense clinical benefit though does not seem possible at this time.

## 8. Conclusions

Necrotizing enterocolitis remains a significant cause of morbidity and mortality in the neonatal population, predominantly affecting premature and low-birth-weight infants. The complex interplay of an underdeveloped gut barrier, immature immune responses, underdeveloped gut flora, and environmental factors creates a precarious environment where oxidative stress (OS) can exacerbate intestinal injury. The role of redox chemistry in NEC is evident, with reactive oxygen species (ROS) and reactive nitrogen species (RNS) contributing to tissue damage and the inflammatory cascade. Despite the body’s intrinsic antioxidant defenses, the overwhelming burden of RS in neonates surpasses these defenses, leading to oxidative damage. Current therapeutic strategies focus on minimizing oxidative stress through the promotion of breastfeeding, supplementation/altering of formulas, judicious use of parenteral nutrition, and goal-oriented use of high-concentration oxygen. Emerging research into antioxidants such as melatonin and advanced diagnostic markers offer hope for better prevention and treatment. Continued exploration into the mechanisms of OS and the development of targeted interventions hold promise for reducing the incidence and severity of NEC and ultimately improving outcomes for vulnerable neonates.

## Figures and Tables

**Figure 1 ijms-25-08416-f001:**
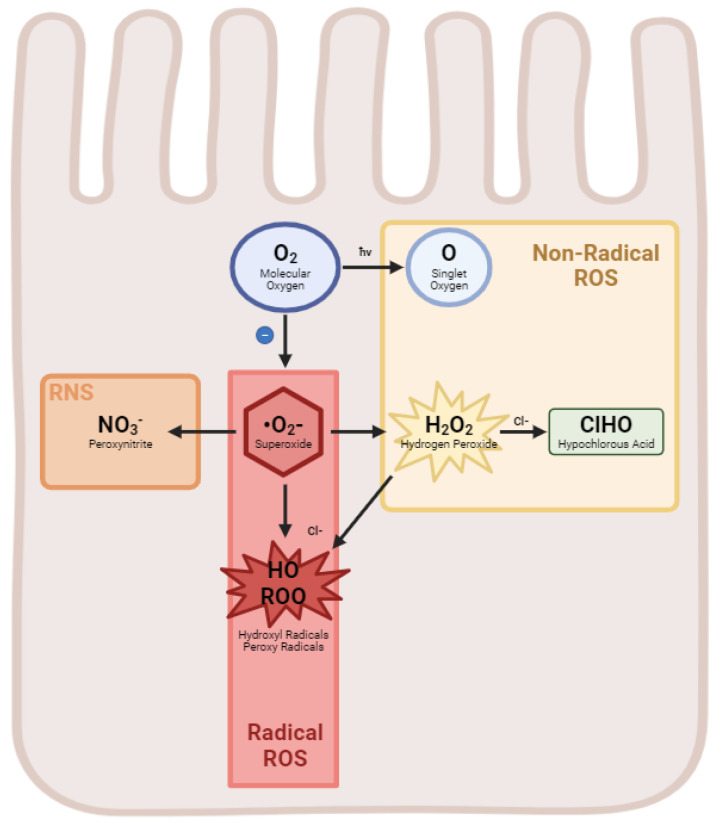
Generation of reactive oxygen species in the enterocyte. A diagram illustrating the flow of molecular oxygen to form various radical and non-radical ROS as well as the RNS Peroxynitrite.

**Figure 2 ijms-25-08416-f002:**
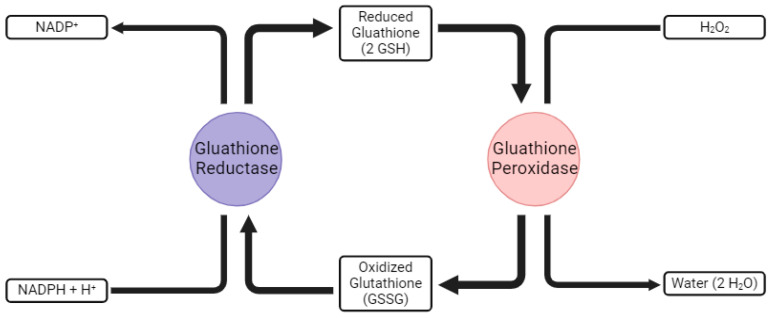
Glutathione recycling. A schematic demonstrating the relationship of glutathione reductase and glutathione peroxidase, as well as the recycling of oxidized glutathione to reduced glutathione.

**Table 1 ijms-25-08416-t001:** A list of commonly encountered reactive oxygen and reactive nitrogen species.

Reactive Oxygen Species	Formula
Hydrogen Peroxide	H_2_O_2_
Superoxide anion	°O_2_^−^
Hydroxyl Radical	HO
Singlet oxygen	O
Ozone	O_3_
Peroxyl radical	ROO
Hypochlorous acid	ClHO
**Reactive Nitrogen Species**	
Nitric oxide	NO
Peroxynitrite anion	NO_3_^−^

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
