# Peer review of "Redox Chemistry: Implications for Necrotizing Enterocolitis"

_ijms, 2024, doi:10.3390/ijms25158416_

Round 1
Reviewer 1 Report
Comments and Suggestions for Authors
The paper write by Hunter and Gershner could be of interest to the scientific. However, in its current form it focus on NEC and not on Redox Chemistry.
So, the title should be changed accordingly. The flow of the paragraphs should also be changed. In particular, the authors should insert the general part of Redox Chemistry after NEC because this paragraph is important for understanding the implications in NEC. Furthermore, a graphic abstract would help the reader to better understand the meaning and information contained in the paper
Reviewer 2 Report
Comments and Suggestions for Authors
This manuscript, from a scientific perspective, is reasonably good/acceptable in terms of its contents and “layout”.
However, there is no doubt that it needs to be improved before being accepted for publication.
11. The title needs to be modified; why not use the following(?):
“Redox Chemistry: Implications for Necrotizing Enterocolitis”
22. What are the hallmarks of necrotizing enterocolitis? These need to be addressed in the introduction in a more thoughtful/rational/sequential/scientific manner.
33. Pro-inflammatory cytokines and chemokines are 'early warning' biomarkers for diagnosing NEC; the foregoing merits a short section on its own in combination with other biomarkers of oxidative stress and how they apply to necrotizing enterocolitis
44. I would encourage the authors to examine and cite the references given below. Some of them are “classical” but still relevant to research in the areas of both redox chemistry and necrotizing enterocolitis. Furthermore, I invite the authors to examine this “collection” of references in order to add/improve the academic content of the manuscript; such that in the future this review garners many more citations in the scientific literature than it would in its current form.
55. There are countless grammatical errors which need to be corrected e.g., line 94: Superoxide to superoxide and line 95; Catalase to catalase.
66. Please check the notation used in the manuscript for oxidized/reduced species, such that it follows IUPAC recommendations.
……………………………………
Redox metabolism: ROS as specific molecular regulators of cell signalling and function
Claudia Lennicke & Helena M. Cochemé, Molecular Cell 81, 18, P3691-3707, 2021
………………………………………………
Barbara E. Corkey and Jude T. Deeney, The Redox Communication Network as a Regulator of Metabolism, Front Physiol. 2020; 11: 567796.
………………………………………………………….
Systems redox biology in health and disease
Martin Feelisch, et al., EXCLI J. 2022; 21: 623–646.
,,,,,,,,,,,,,,,,,,,,,,,,,,,,,,,,,,,,,,,,,,,,,,,,,,,,,,,,,,,,,,,,,
Volume Atala Bihari Jena et al, Cellular Red-Ox system in health and disease: The latest update; Biomedicine & Pharmacotherapy, 162, 2023, 114606
………………………………………………………………
Nigel J Hall and Ian Jones, Contemporary Outcomes for Infants with Necrotizing Enterocolitis—A Systematic Review, The Journal of Pediatrics 220, 2019
……………………………………………………………..
Xue Cai et al., A Review of the Diagnosis and Treatment of Necrotizing Enterocolitis, Current Pediatric Reviews 18(3), 2022
…………………………………………
Diego F. Niño, et al. Necrotizing enterocolitis: new insights into pathogenesis and mechanisms
Nature Reviews Gastroenterology & Hepatology, 13, 590–600 (2016)
……….
Steven X. Cho et al., Nature Communications volume 11, Article number: 5794 (2020)
……………………………………..
Geoanna M. Bautista et al.,State-of-the-art review and update of in vivo models of necrotizing enterocolitis, Front. Pediatr., Volume 11,2023 Sec. Neonatology (one of twelve articles that the authors need to investigate and note some important points from the foregoing manuscripts)
……………………………………………………….
Arianna Aceti et al., Oxidative Stress and Necrotizing Enterocolitis: Pathogenetic Mechanisms, Opportunities for Intervention, and Role of Human Milk, Oxid Med Cell Longev. 2018; 2018: 7397659.
Comments on the Quality of English Language
As stated above there are numerous mistakes in the manuscript relating to grammar/syntax and notation. The foregoing need to be corrected.
Round 2
Reviewer 1 Report
Comments and Suggestions for Authors
The authors answered all my questions. Paper has improved with respect to previous version
Reviewer 2 Report
Comments and Suggestions for Authors
The revised manuscript is now acceptable for publication.